# Unsupervised Multivariate Feature-Based Adaptive Clustering Analysis of Epileptic EEG Signals

**DOI:** 10.3390/brainsci14040342

**Published:** 2024-03-30

**Authors:** Yuxiao Du, Gaoming Li, Min Wu, Feng Chen

**Affiliations:** 1School of Automation, Guangdong University of Technology, Guangzhou 510006, China; yuxiaodu@gdut.edu.cn (Y.D.); 2112204375@mail2.gdut.edu.cn (G.L.); 2School of Automation, China University of Geosciences, Wuhan 430074, China; wumin@cug.edu.cn

**Keywords:** epileptic EEG signals, multivariate features, sparrow search algorithm, adaptive clustering

## Abstract

Supervised classification algorithms for processing epileptic EEG signals rely heavily on the label information of the data, and existing supervised methods cannot effectively solve the problem of analyzing unlabeled epileptic EEG signals. In the traditional unsupervised clustering algorithm, the number of clusters and the global parameters must be predetermined, and the algorithm’s analytical results are combined with a huge number of subjective errors, which affects the detection accuracy. For this reason, this paper proposes an unsupervised multivariate feature adaptive clustering analysis algorithm based on epileptic EEG signals. First, CEEMDAN and CWT are introduced into the epileptic EEG signal after preprocessing for joint denoising to further improve the signal quality. Then, the multivariate feature set of the signal is extracted and constructed, which includes nonlinear, time, frequency, and time-frequency characteristics. To reveal the hidden structures and correlations in the high-dimensional feature data, t-SNE dimensionality reduction is introduced. Finally, the DBSCAN clustering algorithm is optimized using the SSA algorithm to achieve adaptive selection of cluster number and global parameters.It not only enhances the clustering performance and reliability of the clustering results, but also avoids subjective errors in the analysis results. It provides a pre-theoretical foundation for the successful development of future seizure prediction devices and has good application prospects in clinical diagnosis and daily monitoring of patients.

## 1. Introduction

Machine learning-based models for recognizing epileptic EEG signals are becoming an increasingly popular research topic are becoming an increasingly popular research topic. In 2007, the first open EEG database for seizure prediction appeared internationally and an algorithmic competition on seizure prediction was initiated to facilitate the comparison of algorithmic performances and to keep updating them [1]. Shoeb et al. used scalp EEG from the CHB-MIT dataset to apply machine learning to seizure detection and found excellent results [2]. Tiwari et al. classified and identified seizures and seizure-free seizures using a keypoint-based computing local binary pattern (LBP) approach using an Support Vector Machine classifier [3]. Al-Hadeethi et al.to achieve satisfactory results for the Adaptive Boosting-Least Squares-Support Vector Machine (AB-LS-SVM) classification model. First, a covariance matrix is used to lower the dimension of the EEG signal, then its statistical features are extracted, and the set with the most significant features is obtained using a non-parametric test [4]. Vicnesh et al. classified distinct epilepsy types by extracting nonlinear characteristics from EEG data and putting them into a decision tree [5].

From 2015 to the present, deep learning technology is increasingly developing rapidly, and the models of neural networks are widely used in several fields. Zheng et al. introduced an epilepsy prediction approach using Convolutional Neural Network, and the modeling process does not require steps such as signal preprocessing or data conversion [6]. Zhang et al. extracted the scalp EEG signal’s time and frequency domain differentiating characteristics using wavelet packet decomposition and standard spatial pattern methods. The pre-seizure and inter-seizure phases were then classified using a shallow Convolutional Neural Network [7]. Ma et al. applied Recurrent Neural Network with Long Short-Term Memory(LSTM) for the first time in epileptic seizure prediction, where they fed the statistical properties of the acquired EEG data into the LSTM architecture [8]. Daoud et al. created an LSTM-based seizure prediction algorithm to target specific individuals [9]. To identify epileptic seizures from EEG recordings, Jana et al. input the produced spectral graph matrix into a one-dimensional convolutional neural network [10]. Hu et al. present a unique strategy that uses a deep bidirectional long short-term memory network for seizure detection [11]. To enhance seizure prediction performance, Tsiouris et al. built a two-layer LSTM network and four preseizure windows of varying lengths [12].

The existing seizure detection techniques have shown good performance and can accurately classify epileptic seizures from non-epileptic cases. However, since classification algorithms need to use a large amount of data with known labels when training classifiers, that is when using classification algorithms to process epileptic EEG data, it requires a lot of time and manpower to label a huge amount of EEG signals. Therefore, the use of supervised classification algorithms to process EEG signals is not compatible with practical applications and cannot be effectively migrated to the task of analyzing unlabeled EEG signals.To explore unlabeled epileptic EEG signals, Wen et al. built a model that utilized deep convolution and autoencoder to perform unsupervised learning of epileptic EEG signal properties [13]. Giridhar P et al. analyzed epileptic EEG signals of epileptic patients by Fuzzy C-mean cluster analysis method to observe the relationship between seizures and clustering coefficients [14]. Liu et al. extracted various features of epileptic EEG signals for cluster analysis [15]. Wu et al. collected four distinctive characteristics from epileptic EEG data and utilized Fuzzy C-means cluster analysis with cluster center number n [16]. Carolina et al. suggested a detection algorithm that uses both the S-transform and the Gaussian Mixture Model, which first performs the Stockwell transform on EEG signals and extracts features, and then uses the Gaussian Mixture Model to detect and analyze epileptic seizures [17]. Wan et al. combined several signal analysis methods, and the epileptic EEG signals were processed by Stockwell’s positive inverse transform and singular value decomposition, respectively, and four features were extracted and analyzed by clustering with the improved Fuzzy C-means algorithm [18]. Carolina et al. applied the Gaussian Mixture Model clustering method to evaluate the EEG waveforms of pediatric epileptic patients [19].

The importance of the clustering algorithm is to discover the information of the data Without any prior information, however, in classical clustering algorithms, the amount of clusters and algorithm parameters must be predetermined, and the algorithm analysis findings are mixed with a large number of subjective errors, which affects the detection accuracy, the parameters that apply to data sets with different structures and do not rely on the selection of a priori information are very important for the clustering algorithm. For this reason, this paper proposes an unsupervised multivariate feature adaptive clustering analysis algorithm for epileptic EEG signals. Compared with other artificial intelligence methods, the adaptive clustering method, which does not need to train the data before analysis, is more objective when analyzing epileptic EEG signals. In this study, the sparrow search method was used to improve the DBSCAN clustering algorithm by adaptively determining the number of clusters and global parameters, which avoids subjective errors in the analysis results, further enhances clustering performance and trustworthiness, which is helpful for healthcare personnel’s clinical diagnosis and the patient’s daily home monitoring in later stages. The flow chart of the full paper is shown in Figure 1. This paper will describe the research content and method in detail step by step.

## 2. Complete Ensemble Empirical Modal Decomposition of Adaptive Noise

In recent years, TORRES et al. proposed an adaptive noise complete ensemble empirical modal decomposition, also known as complete ensemble empirical modal decomposition (CEEMDAN) [20]. CEEMDAN introduces an adaptive noise-assisted method to better deal with high-frequency noise in signals. CEEMDAN can decompose complex vibration signals into multiple intrinsic modal components (IMFs) related to changes in the sampling frequency and the signal itself, and add the IMF component of the white noise at each decomposition, the additional noise is gradually decreased, and there is less residual noise in the inherent modal components, which significantly decreases the rebuilding error. The CEEMDAN decomposition conducts overall averaging on the first-order IMF components acquired to produce the final first-order IMF components, then repeats the aforesaid procedure on the leftover parts. This efficiently solves the problem of moving white noise from high frequency to low frequency, with a global stopping condition at each level of the decomposition, which makes the computation fast and the decomposition most efficient.

To highlight the superiority of CEEMDAN decomposition, Empirical Mode Decomposition (EMD), Ensemble Empirical Mode Decomposition (EEMD), and CEEMDAN decomposition are used in this paper for comparison. From the EMD decomposition components, the low-frequency component modal aliasing phenomenon can be seen more obviously. The complex frequency domain diagram of EMD signal decomposition is shown in Figure 2. In response to the shortcomings of EMD decomposition, EEMD decomposition adds white noise to the original signal for analysis, which can suppress modal aliasing to a certain extent, but is prone to false modal components, signal distortion, and the difficulty of not being able to eliminate the equal transmission of white noise.The complex frequency domain diagram of EEMD signal decomposition is shown in Figure 3. Based on the shortcomings of the above two methods, the CEEMDAN decomposition method can adaptively add white noise to the original EEG signal, and the method can better eliminate its unequal transmission defect.The complex frequency domain diagram of CEEMDAN signal decomposition is shown in Figure 4.

## 3. Continuous Wavelet Transform

Continuous Wavelet Transform (CWT) is a frequently used signal processing approach based on multi-scale wavelet analysis and frequency selectivity that may be used to eliminate noise. In CWT, the signal undergoes a convolution operation with a series of wavelet functions that are concurrent in both the frequency and time domains. Wavelet functions of different scales can provide sensitivity to different frequency components. By performing CWT on the signal, a plot of the wavelet coefficients of the signal at different scales and frequencies can be obtained. Typically, noise is characterized by a random distribution in the wavelet coefficient maps. To remove the noise, the sparseness of the wavelet coefficients can be utilized. That is, a clear signal usually has only a small number of distinctly non-zero wavelet coefficients, whereas noise produces smaller amplitudes that are more evenly distributed across multiple wavelet coefficients. Therefore, noise can be effectively removed by thresholding the wavelet coefficients by setting coefficients less than some threshold value to zero. To get the optimal denoising impact, the threshold value can be adaptively set based on the characteristics of the signal and noise. The steps are as follows:

Suppose $F\left(t\right)\in L^{2}\left(R\right)$ is a wavelet basis function.
(1)$$\int_{R}\frac{{\left|F\left(w\right)\right|}^{2}}{{\left|w\right|}^{-1}}dw<\infty$$
where: $F\left(w\right)$ is the Fourier transform of.

The continuous wavelet basis function is obtained by telescoping and translating the $F\left(t\right)$ transformation.
(2)$$F_{a,b}\left(t\right)=\frac{1}{\sqrt{\left|a\right|}}F\left(\frac{t-b}{a}\right)$$
where: *a* is the scale factor; *b* is the translation factor.

For any time series, its CWT is: (3)$$y^{\left(t\right)}=y^{\left(t-1\right)}+\gamma \frac{\partial C}{\partial y}+\phi \left(t\right)\left(y^{\left(t-1\right)}-y^{\left(t-2\right)}\right)$$

From Equations (2) and (3), the CWT decomposes the original time series at different scales, each corresponding to a different center frequency, by varying the value of the scale factor *a*.

## 4. t-Distributed Stochastic Neighbor Embedding

t-Distributed Stochastic Neighbor Embedding (t-SNE) is a strong tool for the display and dimensionality reduction of high-dimensional data. This algorithm was first proposed by Maaten and Hinton in 2008 [21]. Unlike other dimensionality reduction algorithms, t-SNE better preserves the local features of the data during the dimensionality reduction process, avoids the loss of important information, and overcomes the limitations that exist when dealing with high-dimensional nonlinear data. t-SNE algorithm’s main contribution is that it uses a probabilistic-based approach to measure the resemblance between high-dimensional points of information and attempts to keep these parallels in the space of low dimensions. t-SNE uses a special probability distribution (t-distribution) to be able to effectively deal with outliers in high-dimensional data and to generate better clustering effects in low-dimensional space, revealing hidden structures and associations in high-dimensional data.

To illustrate the superiority of the t-SNE algorithm, this paper utilizes Principal Component Analysis (PCA), Multidimensional Scaling (MDS), Kernel Principal Component Analysis (KPCA), Isometric Mapping (Isomap), and Singular Value Decomposition (SVD) algorithms for data downscaling visualization of some high-dimensional feature data, respectively. Figure 5 show the two-dimensional and three-dimensional distributions after dimensionality reduction of the algorithms, respectively. As may be observed, when compared to various dimensionality reduction techniques, the t-SNE dimensionality reduction algorithm effectively reveals the hidden structures and associations in the high-dimensional data and produces more meaningful clusters.

## 5. SSA-DBSCAN Clustering Algorithm

### 5.1. Density-Based Spatial Clustering of Applications with Noise

Density-Based Spatial Clustering of Applications With Noise (DBSCAN) is a density-based method of clustering whose clustering results are independent of the original order in which the data items. The advantage of the DBSCAN clustering algorithm is that it automatically determines the number and shape of clusters by the range of the radius Eps and the number of dots MinPts, which do not need to be specified in advance. The algorithm’s central premise is that for every data object in a cluster, the neighborhood of a given Eps must contain at least MinPts data objects. Eps Nearest Neighbors denote the nearest neighbors within the Eps radius of a given object are called the Eps nearest neighbors of that object and are denoted as $NEps\left(p\right)$.
(4)$$NEps\left(p\right)=\left\{q\in D|dist\left(p,q\right)\leq Eps\right\}$$

Direct density reachability means that for a given MinPts and Eps, a direct density reachability from an object *q* to *p* is possible, subject to the following conditions.
(5)$$p\in NEps\left(q\right),\left|NEps\left(q\right)\right|\geq MinPts$$

Although DBSCAN clustering does not need specifying the amount of initial clustering points, it tends to be susceptible to the selection of global parameters Eps and MinPts, which relies on human intervention and is selected entirely based on subjective experience in the process of practical application and lacks adaptivity, which has a great impact on the breadth of the algorithm’s application and the credibility of the clustering results. This paper proposes an SSA-optimized DBSCAN clustering algorithm to automatically obtain Eps and MinPts, accomplish adaptable choice of global parameters and increase the trustworthiness of the clustering findings.

### 5.2. Sparrow Search Algorithm

Sparrow Search Algorithm (SSA) is a novel form of swarm intelligence optimization algorithm that has the advantages of high optimality finding capability and quick convergence speed. The method, introduced by Xue and Shen from Donghua University, primarily replicates the foraging and anti-predation behaviors of sparrow groups [22]. The processes of the method are listed below:Initialization of sparrow population position, fitness and initial values of *N*, *n* and ST parameters (maximum number of iterations *N*, population size *n*, safety value ST, pre-warning values $R_{2}$);Start the loop with iteration<N;The population is sorted to derive the current location of the optimal sparrow individual, and the best fitness value (for the first generation of sparrows, the initial optimum is derived. The optimal individual can prioritize access to food);Foraging behavior, the PN sparrows with the best position in each generation are selected as explorers and the remaining n−PN sparrows as followers. Update the explorer position by the following equation:
(6)$$x_{i,d}^{t+1}=\left\{\begin{array}{c} x_{i,d}^{t}\cdot \exp\left(\frac{-i}{\alpha \cdot iter_{\max}}\right),R_{2}<ST\\ x_{i,d}^{t}+Q,\;\;\;R_{2}\geq ST \end{array}\right.$$Equation $x_{i,d}^{t+1}$ denotes the dth dimensional position of the ith sparrow in generation t in the population. α is a uniform random number in (0, 1]. *Q* is a standard normally distributed random number. ST is the warning threshold, which takes the value in the range of (0.5, 1.0]. $R_{2}$ is a uniform random number in (0, 1]. When $R_{2}$ is larger than ST, the explorer moves randomly to the neighborhood of the current position according to the normal distribution, and its value converges to the optimal position;Update the follower position according to the following formula:
(7)$$x_{i,d}^{t+1}=\left\{\begin{array}{c} Q\cdot \exp\left(\frac{xw_{i,d}^{t}-x_{i,d}^{t}}{\alpha \cdot iter_{\max}}\right),i>n/2\\ xb_{i,d}^{t}+\frac{1}{D}\sum_{d=1}^{D}\left(\mathrm{rand}\left\{-11\right\}\right)\cdot \left(\left|xb_{i,d}^{t}-x_{i,d}^{t}\right|\right),i\leq n/2 \end{array}\right.$$The equation xw denotes the position of the worst-positioned sparrow in the population. xb denotes the position of the optimally positioned sparrow in the population. When i>n/2 is used, the function value is the product of a standard normally distributed random number and an exponential function with a natural logarithmic base. When the population converges, the value is consistent with a standard normally distributed random number. When i≤n/2 is used, the function value is the current optimal sparrow location plus a random addition or subtraction of that sparrow’s distance from the optimal location in each dimension, dividing the sum equally into each dimension;Anti-predation behavior to update sparrow population locations:
(8)$$x_{i,d}^{t+1}=\left\{\begin{array}{c} xb_{i,d}^{t}+\beta \cdot \left(x_{i,d}^{t}-xb_{i,d}^{t}\right),f_{i}\neq f_{g}\\ x_{i,d}^{t}+K\cdot \left(\frac{x_{i,d}^{t}-xw_{i,d}^{t}}{\left|f_{i}-f_{w}\right|+\varepsilon }\right),f_{i}=f_{g} \end{array}\right.$$When a sparrow population forages for food, individuals in the population are simultaneously alert to their surroundings. When danger is detected, both explorers and followers abandon the food and move to a new location. In the formula, the absolute value of the denominator is increased to prevent the denominator from taking the value 0. β is a random number that conforms to the standard normal distribution. *k* is a uniform random number of [−1,1]. To prevent the denominator from being unique ε is a smaller number. $f_{w}$ is the fitness value of the sparrow in the worst position;Update the historical optimal fitness;When the maximum number of iterations has been achieved, complete steps 3–7 and exit the loop. Produce the ideal individual posture and fitness value.

### 5.3. SSA-DBSCAN

To overcome the primary shortcoming of the DBSCAN algorithm: the global parameters Eps and MinPts are dependent on manual empirical selection, which is a heavy and unreasonable workload. In this paper, the SSA algorithm is introduced based on DBSCAN’s clustering algorithm, the contour coefficient is used as a measure of the fitness function, and the labels of the clusters are recorded in each optimization to realize the adaptive selection of the global parameters, which further improves the clustering effect. The flow of the SSA-DBSCAN clustering algorithm is shown in the Figure 6.

Initialization of algorithm parameters. Initialize parameters such as the maximum number of sparrow iterations and population size, set the range of global parameters, and randomly generate the initial position of sparrows;Calculate the individual fitness value of the sparrow flock. Calculate the individual fitness value of the sparrow flock according to the objective function equation, and obtain the optimal value of the individual and the group by comparison;Update the position of the individual sparrow. Determine the position of the sparrow according to the warning value of the individual sparrow. Update the explorer position according to Equation (Equation 6) and update the follower position according to Equation (Equation 7). Calculate the individual adaptation value after updating the position of the sparrow flock, sort all the individual adaptation values, and record and save the well-adapted individuals;The anti-predation behavior of sparrows generates a new population, updates the position of the sparrow population according to Equation (Equation 8), and calculates the SC profile coefficients to update the historical optimal fitness values based on the labels obtained from clustering;Determine the relationship between the maximum number of iterations and the current number of iterations, when the maximum number of iterations is less than the current number of iterations, the search for the optimal end, the output of the optimal global parameters Eps and MinPts, and get the corresponding clustering results, otherwise return to step 2.

## 6. Experimental Results and Analysis

### 6.1. EEG Data

The dataset in this paper is a sample segmented EEG time series recordings of ten epileptic patients collected from the Neurology and Sleep Center, Hauz Khas, New Delhi [23]. Patients were placed scalp EEG electrodes according to a 10–20 electrode system with a sampling rate of 200 HZ. The signal is filtered at 0.5–70 Hz and divided into three distinct periods. Each stage contains a MAT file of 50 EEG time series signals with a time length of 5.12 s. During artificial cropping, cardiac EMG and some industrial frequency interferences were removed, so no preprocessing of the signals was required. The representative signals of each stage are shown in the Figure 7: it can be seen that there exists a portion of waveforms with small differences, and it is easy to make a misjudgment only by the naked eye.

### 6.2. Joint Denoising

Raw epileptic EEG signals still inevitably have some noisy data even after manual processing, and this paper further improves the signal quality by introducing signal decomposition. The CEEMDAN algorithm enhances the stability of the EMD method by adding a noise adjustment parameter when performing signal decomposition. However, this method is more sensitive to noise, and the decomposition results of the CEEMDAN algorithm may be greatly affected when the noise level is high. Especially for non-smooth signals or signals containing more noise, the decomposition results of the CEEMDAN algorithm may deteriorate. Therefore, in practical applications, the decomposed signals need to be processed appropriately. On the one hand, after the EEG signal is decomposed by CEEMDAN, each IMF component is arranged from high to low according to the instantaneous frequency, but it is difficult to select the optimal IMF component from all the components. As a result, the correlation coefficient, which measures the degree of correlation between the original signal and each component, is used as a criterion for selecting the best IMF component. The correlation coefficient is computed as follows: (9)$$\rho =\frac{\sum_{t=1}^{n}\left(x\left(t\right)-\bar{x}\right)\left(f_{IMF_{K}}\left(t\right)-{\bar{f}}_{IMF_{K}}\right)}{\sqrt{\sum_{t=1}^{n}{\left(x\left(t\right)-\bar{x}\right)}^{2}}\sqrt{\sum_{t=1}^{n}{\left(f_{IMF_{K}}\left(t\right)-{\bar{f}}_{IMF_{K}}\right)}^{2}}}$$
where $\bar{x}$ is the average value of $x\left(t\right)$, ${\bar{f}}_{IMF_{K}}$ is the average value of $f_{IMF_{K}}\left(t\right)$.

On the other hand, some IMF components, especially high-frequency IMF components, often contain both signal and noise components, and directly discarding that element of the IMF component increases the danger of losing the effective signal while eliminating noise.. Therefore, the reconstructed signal after CEEMDAN decomposition and screening is processed by CWT, which can decompose it in the frequency domain and obtain the sub-time series on different scales. According to the characteristics of multi-scale analysis and frequency selectivity of CWT, the sparsity of wavelet coefficients is utilized to remove the noise from the signal, thus further improving the signal quality.The CEEMDAN decomposition findings are displayed in Figure 8.

The CEEMDAN decomposition technique is used to produce numerous IMF components from epileptic EEG data, and the correlation coefficients between each IMF component and the original EEG signal are obtained. The correlation coefficient will be low when the noise content in the IMF components is high. In this paper, We decide to eliminate the IMF components with correlation coefficients less than 0.3 and recreate the signal of the remaining high correlation IMF components, and use the db5 wavelet basis function for CWT processing after reconstruction. The correlation coefficients of each IMF component are shown in the Table 1. The results of CWT processing are shown in the Figure 9.

To analyze the experimental results qualitatively and quantitatively, the evaluation indexes of the denoising effect are introduced: signal-to-noise ratio (SNR), root mean square error (RMSE), normalized correlation coefficient (NCC), and peak signal-to-noise ratio (PSNR). The larger the SNR after denoising, the better the denoising effect is. RMSE represents the square root of the variance between the initial signal and the denoised signal, the smaller the value, the higher the measurement accuracy. NCC reflects the difference between the waveforms of the initial signal and the denoised signal, the smaller the difference is, the better the denoising effect is proved. The larger the PSNR is, the more feature peak information is retained after denoising, and the better the denoising effect is.

The signal-to-noise ratio equation is as follows: (10)$$\mathrm{SNR}=20\mathrm{lg}\frac{\sum_{t=1}^{N}x^{2}\left(t\right)}{\sum_{t=1}^{N}{\left[x\left(t\right)-\overset{*}{x\left(t\right)}\right]}^{2}}$$

The waveform similarity formula is as follows: (11)$$\mathrm{NCC}=\frac{\sum_{t=1}^{N}x\left(t\right)\overset{*}{x\left(t\right)}}{\sqrt{\sum_{t=1}^{N}x^{2}\left(t\right)\sum_{t=1}^{N}\overset{*}{x{\left(t\right)}^{2}}}}$$

The formula for the root mean square error is as follows: (12)$$\mathrm{RMSE}=\sqrt{\frac{1}{N}\sum_{t=1}^{N}{\left[x\left(t\right)-\overset{*}{x\left(t\right)}\right]}^{2}}$$

The formula for the peak signal-to-noise ratio is as follows: (13)$$\mathrm{PSNR}=20\mathrm{lg}\frac{x^{2}{\left(t\right)}_{\max}\times N}{\sum_{t=1}^{N}{\left[x\left(t\right)-\overset{*}{x\left(t\right)}\right]}^{2}}$$

The Table 2 displays metrics like the ratio of signal to noise after joint and single noise reduction. The comparison reveals that CEEMDAN joint CWT denoising is the most effective, with the greatest ratio of signal to noise and the least error in root mean square, the smallest difference in the similarity between the waveform and the reconstructed signal, and the greatest peak ratio between signal and noise.

### 6.3. Feature Dimension Reduction and Clustering

Currently, the major approaches for analyzing epileptic EEG data are time-domain, frequency-domain, and time-frequency-domain analysis. However, It is not feasible to thoroughly evaluate epileptic EEG data from either the time domain or the frequency domain perspective. Only by combining various aspects of epileptic EEG data at the same time can we obtain a more comprehensive study of EEG signals. Twelve characteristics were extracted to reflect the original epileptic EEG data’ characteristics, including Mean, Skewness, Shannon Entropy, Mean Teager Energy, Hjorth parameter, Fluctuation index, and Root mean square, including time, frequency, time-frequency, and nonlinear characteristics. By introducing t-SNE dimensionality reduction translates the highly dimensional characteristic data to a space with few dimensions while keeping the critical information in the data. This not only allows for improved data analysis and processing, but additionally decreases the complexity of computation and enhances the efficiency of the clustering algorithms. The DBSCAN clustering algorithm does not require a predetermined amount of clusters, and using the SSA algorithm to achieve adaptive adjustment of global parameters enhances the reliability of the clustering findings.

This study uses the assessment metrics Silhouette Coefficient (SC), Calinski-Harbasz Score (CH), and Davies-Boulding (DBI) to assess the effectiveness of the suggested clustering algorithm. These indicators are defined below: (14)$$\mathrm{SC}=\frac{b-a}{\max\left(a,b\right)}$$
where *a* represents the average distance between the current sample point and other sample points of the same class, and *b* represents the mean distance between the current sample point and the closest other sample point of another class. A collection’s contour coefficient is the average of all samples’ contour coefficients. The contour coefficient has the following range of values: [−1, 1], and the closer the examples of the same class are near each other, and the farther apart the samples of different classes are from each other, the higher the score.
(15)$$\mathrm{CH}=\frac{tr\left(B_{k}\right)\left(n_{E}-k\right)}{tr\left(W_{k}\right)\left(k-1\right)}$$
where $n_{E}$ is the number of samples and *k* is the number of categories. $B_{k}$ is the covariance matrix between categories, $W_{k}$ is the covariance matrix of internal data, and tr is the trace of the matrix. In simple terms, the lower the covariance of the data inside the categories, the bigger the covariance across the categories, and the higher the CH score, the stronger the clustering effect.
(16)$$\mathrm{DBI}=\frac{1}{N}\sum_{i=1}^{N}\max_{j\neq i}\frac{{\bar{S}}_{i}+{\bar{S}}_{j}}{{∥w_{i}-w_{j}∥}_{2}}$$
where ${\bar{S}}_{i}$ is the average Euclidean distance from the samples of the *i* class to its class center, and ${∥w_{i}-w_{j}∥}_{2}$ is the Euclidean distance from the class centers of the *i* and *j* classes. The lower the value of DBI, the lesser the level of dispersion and the higher the categorization outcome.

To reflect the application value of the SSA-DBSCAN algorithm in processing unlabeled epileptic EEG signals, some traditional clustering algorithms are selected for clustering comparison experiments in this paper. Traditional methods of clustering need a predetermined number of clusters, and this research divides them into three groups based on the composition of the the information set, and the particular findings are displayed in the Figure 10 and the Table 3.

### 6.4. Comprehensive Performance Evaluation

Most of the current epilepsy studies use a single metric to measure the performance of the algorithm, such as accuracy, sensitivity, false prediction rate, etc., and only a few studies have confirmed the significance and rigor of their results in a statistically significant way. To provide a complete evaluation of algorithm performance measures, this work introduces the coefficient of variation technique. The coefficient of variation technique assigns weights by discovering patterns in the data itself. The method uses the degree of variation of the indicators of the evaluated object to determine the weight of the indicators, which can realize the dynamic assignment of the indicators of the evaluated object. The large degree of variation of the indicator indicates that it is more important in the evaluation of the object indicators, and is given a larger weight; conversely, it is given a smaller weight. In this paper, the Silhouette Coefficient (SC) and Calinski-Harbasz Score (CH) are set as positive indicators, Davies-Boulding (DBI) is established as a negative indicator, and the bigger the number, the lower the score, and the way of using the inverse approach to positively normalize the negative indicators. The steps are shown below:

Assuming that the normalization and standardization processed constitute the data matrix $R={\left(r_{ij}\right)}_{m\times n}$, the mean value of the indicator is calculated: (17)$$A_{j}=\frac{1}{n}\sum_{i=1}^{m}r_{ij}$$

Calculate the standard deviation of the indicator: (18)$$S_{j}=\sqrt{\frac{1}{n}\sum_{i=1}^{m}{\left(r_{ij}-A\right)}^{2}}$$

Calculate the coefficient of variation: (19)$$V_{j}=S_{j}/A_{j}$$

Calculate the weights: (20)$$W_{j}=\frac{V_{j}}{\sum_{j=1}^{n}V_{j}}$$

Calculate the score: (21)$$Score_{j}=\sum_{j=1}^{n}W_{j}r_{ij}$$

The final scoring results of the coefficient of variation method are shown in the Figure 11. As can be seen, the method in this article received the highest score of 0.67969, and the score of the DBSCAN algorithm is 0.50883, which is significantly improved after optimization. the GMM algorithm performs poorly. Because both the ISODATA and K-medoids methods are based on the K-means algorithm, their scores are comparable and superior than the K-means algorithm.

### 6.5. Generalizability Analysis

An ideal epileptic EEG signal recognition algorithm should have strong generalization ability and universality. To evaluate the generalizability and efficacy of the technique in this research, the epilepsy dataset from the University of Bonn (Germany) was selected as the generalisation experimental data. The Bonn dataset was published in 2001 by Andrzejak RG, which is an intracranial epilepsy dataset [24]. It comprises of five datasets, Z, O, N, F, and S, with Z and O representing the EEG signals of healthy participants in the cephalic cortex with eyes open and closed, respectively, and the subjects were required to be in the stage of awake and self-relaxation in the acquisition process; F and S are the electrical activity recordings of the epileptogenic region in the focal area during the interictal and ictal phases, respectively; and Category N is the recording of EEG signals from patients located in the intracranial hippocampus structure in the interictal phases. In this paper, a triple categorization task (Z-S-F, O-S-F, Z-S-N) of normal EEG signals, and EEG signals during seizure and interictal periods was designed using the Bonn dataset.

This work introduces the original DBSCAN method for comparative tests in order to properly assess the generalization performance of the improved clustering algorithm. The particular findings are presented in the Table 4. Based on the experimental findings, it can be seen that both algorithms realize effective clustering, and the ultimate number of clusters aligns with the dataset’s class division. The algorithm shown in this articleis better than the DBSCAN algorithm in all indicators due to the introduction of global parameter adaptive selection, and the generalization ability and universality have been further improved.

## 7. Conclusions

Unsupervised multivariate feature-based adaptive clustering analysis algorithm for epileptic EEG signals has superior performance and more objective outcomes in the processing of epileptic EEG data. In this paper, we integrate the multivariate features of epileptic EEG signals to realize a more comprehensive EEG signal analysis. The SSA-DBSCAN clustering model can discover the information of the dataset without any a priori information, and adaptively select the global parameters and the number of clusters to avoid subjective error, to improve the performance of the clustering and the credibility of the clustering results. The final testing findings revealed that the indexes of SC, CH, and DBI reached 0.6775, 4615.198, and 0.53475, respectively. Instead of utilizing a single index to quantify the performance of the method, this study presents the coefficient of variation approach to examine the experimental findings, which confirms the rigor of the results in a statistical sense. The subsequent plan of this paper is to statistically analyze the test results of patients of different genders and age groups, to provide a pre-theoretical basis for the wide application of actual epilepsy clinical assisted diagnostic technology, and to provide reliable algorithmic validation for the development of convenient epilepsy detection devices.

## 8. Discussion

The brain, as an extremely important organ in the human body, is the most complex and advanced part of the central nervous system. Epilepsy, as a disease directly related to the brain, is very difficult to deal with and has no cure for the time being. Epilepsy not just causes bodily agony for sufferers, but it also aggravates their mental load and may easily develop to other ailments. In clinical practice, the diagnosis of epilepsy mainly relies on the judgment of experienced doctors, but the manual judgment will consume a lot of time and energy of experienced doctors on the one hand, and on the other hand, it is prone to subjective differences due to the experience and experience of different doctors. In this paper, we propose an unsupervised multivariate feature adaptive clustering analysis of epileptic EEG signals, but there are some limitations in this study and worthy of future research:This research uses the same extraction of features approach for EEG recordings in various times of epileptic episodes. In the future, We should investigate various extraction of features approaches for different durations of epileptic EEG recordings, and multi-dimensional features should be fused into the model effectively to achieve a better and more stable recognition rate for each period.This paper’s categorization of pre-seizure, inter-seizure, and post-seizure phases is based on the experience of previous researchers. Since each epileptic patient has different physical characteristics, seizure type and reaction time, we need to develop an adaptive classification method according to the patient’s own characteristics in order to accurately predict each epileptic patient in the future.Due to the limitations of the experimental conditions, this study only achieved good results on the public data set, and whether this algorithm can be applied to epileptic patients of all ages needs to be verified. Whether the algorithm in this study meets the needs of clinical treatment remains to be verified.In terms of hardware implementation, because of the intricacy of signals from the EEG, deploying the algorithm to run on a hardware platform requires consideration of hardware-software co-design, and the model can be subsequently lightened to meet the clinical demand for efficient online epilepsy detection on low-power hardware systems.

## Figures and Tables

**Figure 1 brainsci-14-00342-f001:**
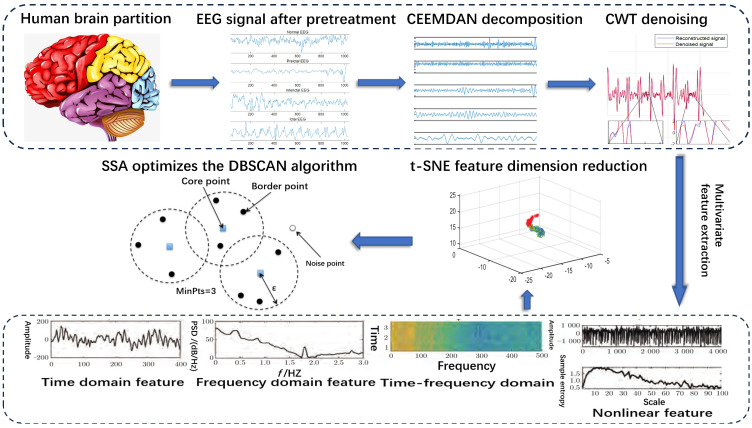
Full text flow chart.

**Figure 2 brainsci-14-00342-f002:**
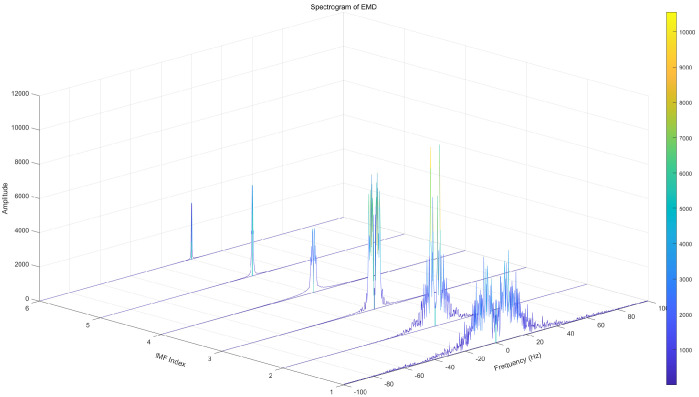
Complex frequency domain diagram of EMD signal decomposition.

**Figure 3 brainsci-14-00342-f003:**
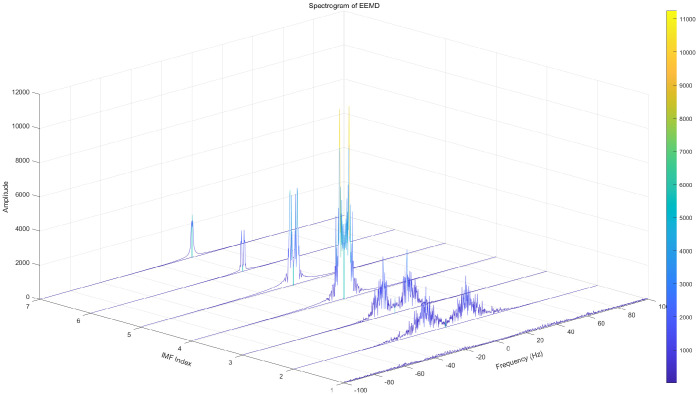
Complex frequency domain diagram of EEMD signal decomposition.

**Figure 4 brainsci-14-00342-f004:**
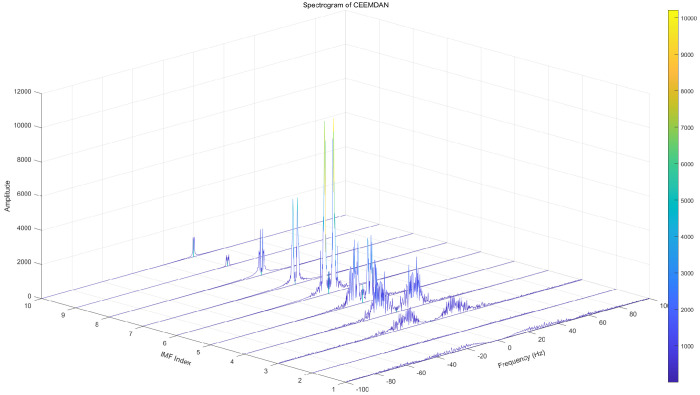
Complex frequency domain diagram of CEEMDAN signal decomposition.

**Figure 5 brainsci-14-00342-f005:**
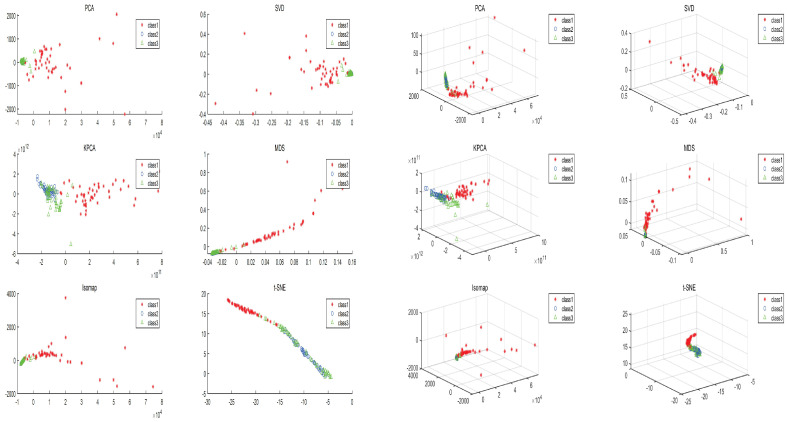
Two-dimensional and Three-dimensional distribution after dimensionality reduction.

**Figure 6 brainsci-14-00342-f006:**
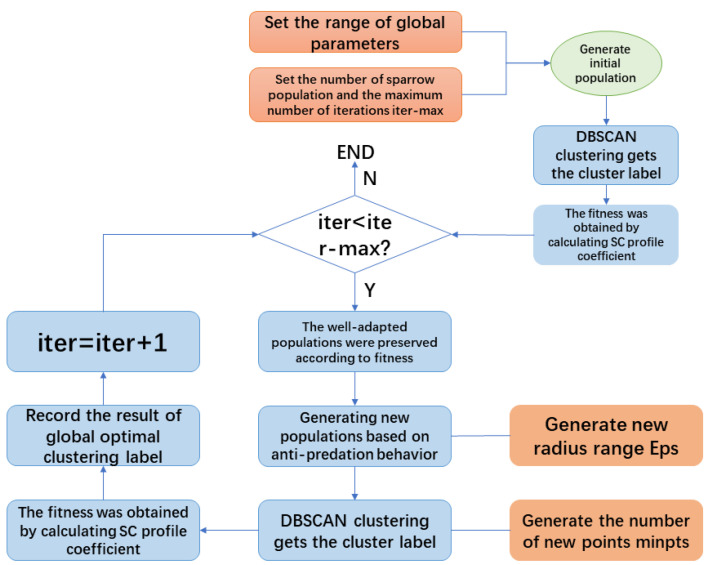
Flowchart of the SSA-DBSCAN algorithm.

**Figure 7 brainsci-14-00342-f007:**
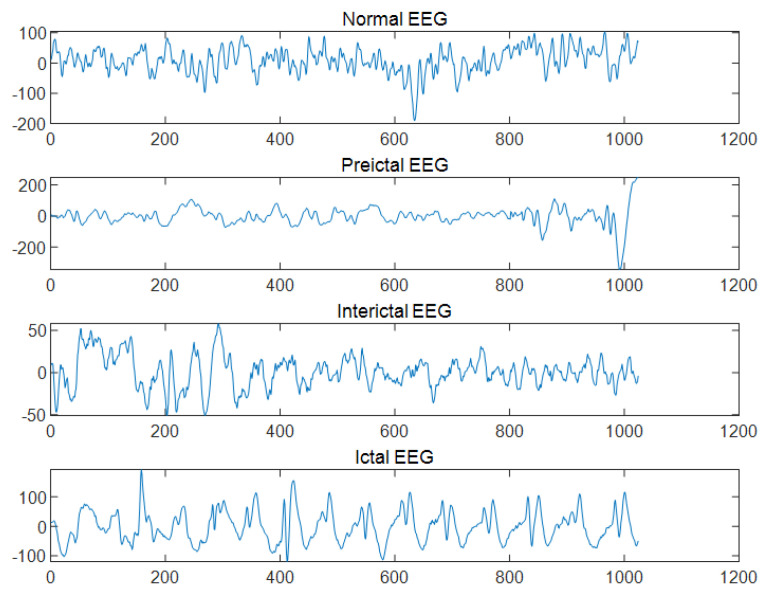
Different stages of epileptic EEG signals.

**Figure 8 brainsci-14-00342-f008:**
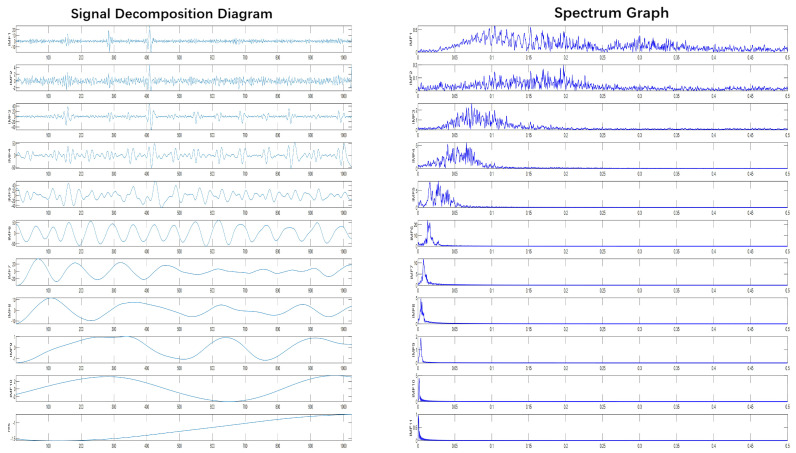
IMF component obtained by CEEMDAN decomposition and its corresponding spectral graph.

**Figure 9 brainsci-14-00342-f009:**
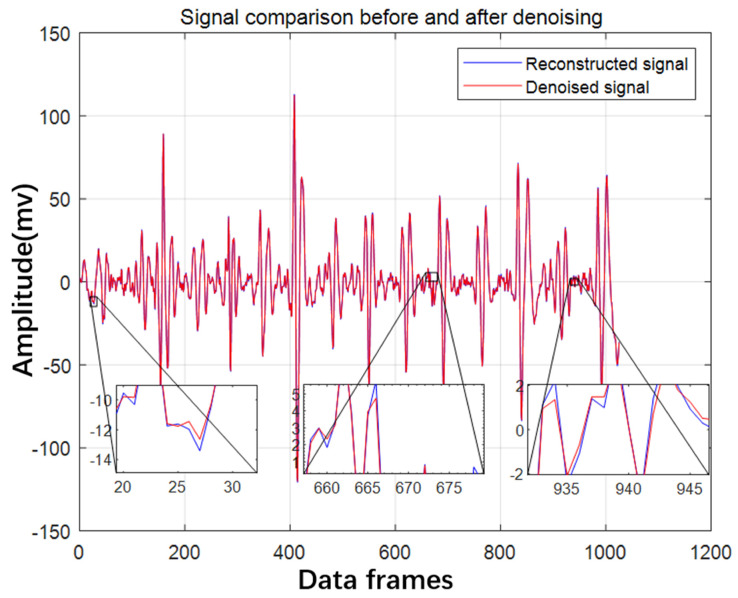
The result of CWT joint denoising.

**Figure 10 brainsci-14-00342-f010:**
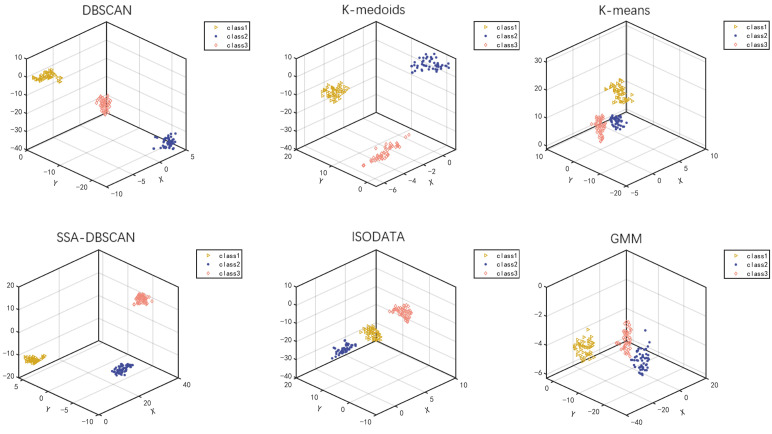
Clustering results of different algorithms.

**Figure 11 brainsci-14-00342-f011:**
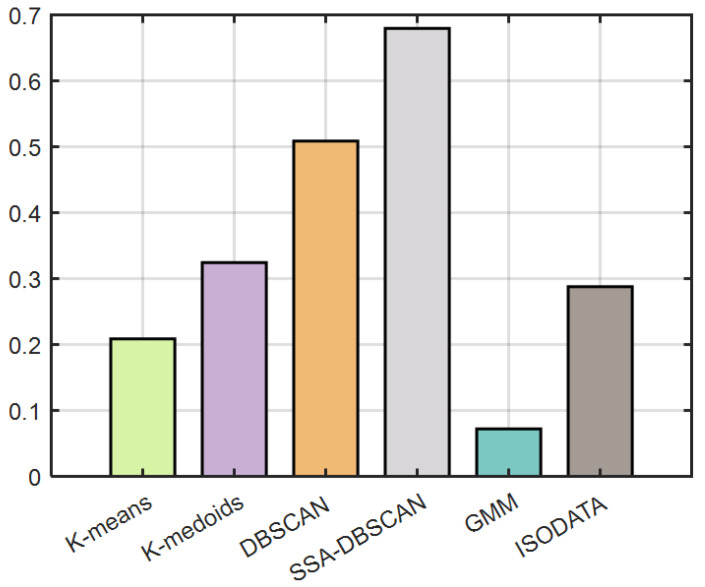
The results of comprehensive evaluation by coefficient of variation method.

**Table 1 brainsci-14-00342-t001:** Correlation coefficient for each IMF component.

IMFs	IMF1	IMF2	IMF3	IMF4	IMF5	IMF6	IMF7	IMF8	IMF9	IMF10
Data	0.1698	0.1525	0.3884	0.5431	0.6904	0.7746	0.3233	0.0466	0.0235	0.0193

**Table 2 brainsci-14-00342-t002:** Comparison of noise reduction effect of different methods.

Method	SNR/dB	RMSE	NCC	PSNR
CEEMDAN	25.7211	0.2872	0.99429	39.5367
CWT	25.0639	0.3261	0.98965	39.2795
CEEMDAN + CWT	26.1206	0.2216	0.99987	40.4548

**Table 3 brainsci-14-00342-t003:** Different algorithm clustering results evaluation index data.

Algorithm	SC	CH	DBI
GMM	0.5822	2242.194	0.81623
K-means	0.6464	3032.312	0.75618
K-medoids	0.6277	3863.054	0.70741
ISODATA	0.6101	3432.406	0.72068
DBSCAN	0.6318	4226.564	0.61713
SSA-DBSCAN	0.6775	4615.198	0.53475

**Table 4 brainsci-14-00342-t004:** Evaluation index data of different classification tasks.

Group	Algorithm	SC	CH	DBI	Categories
Z-S-F	DBSCAN	0.6325	4209.317	0.61938	3
	SSA-DBSCAN	0.6681	4583.274	0.54186	3
O-S-F	DBSCAN	0.6297	4231.462	0.60865	3
	SSA-DBSCAN	0.6712	4607.291	0.53169	3
Z-S-N	DBSCAN	0.6306	4217.614	0.61437	3
	SSA-DBSCAN	0.6659	4592.863	0.53862	3

## Data Availability

The database used in this study is publicly available at websites: https://www.researchgate.net/publication/308719109_EEG_Epilepsy_Datasets, accessed on 30 March 2024.

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
