# Peer review of "Unsupervised Multivariate Feature-Based Adaptive Clustering Analysis of Epileptic EEG Signals"

_brainsci, 2024, doi:10.3390/brainsci14040342_

Round 1

Reviewer 1 Report

Comments and Suggestions for Authors

The development of EEG over decades has been accompanied by the development of statistical analysis methods. This was especially evident in connection with the development of artificial intelligence methods. The manuscript attempts to optimize the DBSCAN clustering algorithm by adding a sparrow search algorithm to implement adaptive selection of global parameters and the number of clusters of the clustering algorithm. In general, the problem raised in the manuscript is relevant and necessary for both clinical diagnosis and scientific research. At the same time, the work requires making a number of changes, incl. restructuring:

1. Introduction - ? Did the authors mark the Introduction with the number zero? At the end of the introduction, perhaps this last block should be rearranged to reflect the purpose of this study. The noted literature review shows a variety of methods for assessing EEG, which is optimally completed by setting a goal.

2. Chapters 1-4 follow. It is not clear what the authors put into the description of these Chapters? This is a continuation of the literature review or description of research methods. These Chapters need restructuring. The Research Methods chapter is needed. The link Data Availability Statement: Data supporting the results reported in this article at https://doi.org/ 10.13140/RG.2.2.14280.32527006 does not open. Source 23 opens well.

3. Figures 1 and 6 are very small. Captions on figures should be enlarged in a graphics editor.

4. Figures 2 and 3 also leave much to be desired.

5. Chapter Discussion – missing. Maybe the authors forgot to insert information here? Moreover, the literature review and results look good.

6. There seems to be some contradiction here: Funding: This research received no external funding Acknowledgments: This work was supported by the Foundation of National Natural Science Foundation of China (Grant No. 61640213 and Grant No. 61976059). Obviously, the link to the grant must be transferred to Funding.

7. Link 14 presents only an abstract from the IFCN congress without serious results. 14. Kalamangalam, G.P.; Chelaru, M. F125. Brain connectivity related to sleep-wake state: An intracranial EEG study. Clinical Neurophysiology 2018, 129, e114.

Perhaps it makes sense to look at the federation's (IFCN) guidelines for analyzing epi-EEG?

Author Response

We gratefully thank editor and all reviewers for their time spend making their constructive remarks and useful suggestions, which has significantly raised the quality of the manuscript and has enables us to improve the manuscript. Each suggested revision and comment, brought forward by the reviewers was accurately incorporated and considered. Below the comments of the reviewers are response point by point and the revisions are indicated. And the full text has been revised carefully for another time.

Reviewer 2 Report

Comments and Suggestions for Authors

An unsupervised multivariate feature adaptive clustering based on epileptic EEG signals was proposed. Literature search and analysis is good to be shown. However, it is hard to recognize what authors want to show in the proposed idea. In addition, it is also hard to understand the effectiveness of the proposed idea. Therefore, there are some comments below.

1. From iThenticate report, the percent match (34%) needs to be lower so authors had better check that.

2. Abbrevivated journal names need to be used in the Reference section.

3. In Figure 1, x and y-label fonts are too small.

4. Figures 2 and 3 could be combined.

5. What are differences in Figure 6 ? There are several graphs with different colors.

6. No contents in 7. Discussion ?

7. Authors mentioned that there is no funding. But there is a funding in Acknowledgments.

8. Why generalizability analysis needs ? In Table 4, there are no too much differences between SC, CH, DBI in same categories.

9. In Figure 8, after SSA-DBSCAN, some data seems to be missing.

10. in Table 2, CEEMDAN+CWT method does not improve SNR, RMSE, NCC, and PSNR compared to CEEMDAN or CWT methods. Why these results are shown ?

10. In Figure 7, there are no too much differences between reconstructed and denoised signals. Why is it ?

11. In Figure 7, what are units of Amplitude and wavelength ?

12. What is the limitation of the proposed work ?

13. In Abstract or Conclusion Sections, valuable simulated data needs to be provided.

14. In Supplementary Materials, the link does not work properly.

15. What is the unit of y-axis in Figure 9 ? The maximum value looks like 1.0.

Author Response

(The authors gave the same response as above.)

Reviewer 3 Report

Comments and Suggestions for Authors

The manuscript in question is undoubtedly relevant and of scientific interest. However, the authors need to make some changes to make the manuscript more readable for experts in various fields. There are many abbreviations in the manuscript; authors must provide a transcript of each. It also makes sense to structure the manuscript more clearly by adding the classic “Materials and Methods” section, which should describe the research methodology in detail, possibly with the use of drawings. In the "discussion" section there is only a link to additional materials; It makes sense for the authors to reorganize this section to formulate a discussion of their results. A link to additional materials can be transferred to the bibliography.

Author Response

(The authors gave the same response as above.)

Round 2

Reviewer 1 Report

Comments and Suggestions for Authors

The paper may be published in its present form.

Reviewer 2 Report

Comments and Suggestions for Authors

Authors carefully revised the manuscript according to the questions so I want to recommend this article could be accepted.